# Impact of Viral Co-Detection on the Within-Host Viral Diversity of Influenza Patients

**DOI:** 10.3390/v17020152

**Published:** 2025-01-23

**Authors:** Su Myat Han, Yoshiano Kubo, Alexis Robert, Marc Baguelin, Koya Ariyoshi

**Affiliations:** 1School of Tropical Medicine and Global Health, Nagasaki University, Nagasaki 852-8523, Japan; koya.ariyoshi@gmail.com; 2Department of Infectious Disease Epidemiology, Faculty of Epidemiology and Population Health, London School of Hygiene and Tropical Medicine, London WC1E 7HT, UK; Alexis.Robert@lsthm.ac.uk (A.R.); marc.baguelin@imperial.ac.uk (M.B.); 3Graduate School of Biomedical Sciences, Nagasaki University, Nagasaki 852-8520, Japan; yoshinao@nagasaki-u.ac.jp; 4Department of Clinical Medicine, Institute of Tropical Medicine, Nagasaki University, Nagasaki 852-8102, Japan; 5Infectious Disease Epidemiology and Dynamics, Institute of Tropical Medicine, Nagasaki University, Nagasaki 852-8102, Japan; 6MRC Centre for Global Infectious Disease Analysis, and the Abdul Latif Jameel Institute for Disease, Imperial College London, London SW7 2AZ, UK

**Keywords:** co-detection or co-infection, influenza, A/H3N2

## Abstract

Numerous studies have documented the evidence of virus–virus interactions at the population, host, and cellular levels. However, the impact of these interactions on the within-host diversity of influenza viral populations remains unexplored. Our study identified 13 respiratory viral pathogens from the nasopharyngeal swab samples (NPSs) of influenza-like-illness (ILI) patients during the 2012/13 influenza season using multiplex RT-PCR. Subsequent next-generation sequencing (NGS) of RT-PCR-confirmed influenza A infections revealed all samples as subtype A/H3N2. Out of the 2305 samples tested, 538 (23.3%) were positive for the influenza A virus (IAV), while rhinovirus (RV) and adenoviruses (Adv) were detected in 264 (11.5%) and 44 (1.9%) samples, respectively. Among these, the co-detection of more than one virus was observed in ninety-six samples, and five samples showed co-detections involving more than two viruses. The most frequent viral co-detection was IAV–RV, identified in 48 out of the 96 co-detection cases. Of the total samples, 150 were processed for whole-genome sequencing (WGS), and 132 met the criteria for intra-host single-nucleotide variant (iSNV) calling. Across the genome, 397 unique iSNVs were identified, with most samples containing fewer than five iSNVs at frequencies below 10%. Seven samples had no detectable iSNVs. Notably, the majority of iSNVs (86%) were unique and rarely shared across samples. We conducted a negative binomial regression analysis to examine factors associated with the number of iSNVs detected within hosts. Two age groups—elderly individuals (>64 years old) and school-aged children (6–18 years old)—were significantly associated with higher iSNV counts, with incidence rate ratios (IRR) of 1.80 (95% confidence interval [CI]: 1.09–3.06) and 1.38 (95% CI: 1.01–1.90), respectively. Our findings suggest a minor or negligible contribution of these viral co-detections to the evolution of influenza viruses. However, the data available in this study may not be exhaustive, warranting further, more in-depth investigations to conclusively determine the impact of virus–virus interactions on influenza virus genetic diversity.

## 1. Introduction

Influenza virus is a negative-sense, segmented, single-stranded RNA virus under the Orthomyxoviridae family, with an approximately 13.5-kilobase (kb) genome [1]. Influenza A viruses (subtype: A/H3N2, A/H1N1) and influenza B viruses (lineages: Victoria and Yamagata) co-circulate among the human population [2], leading to the annual seasonal influenza epidemic worldwide. These seasonal influenza outbreaks remain a major public health threat, affecting 3–5 million people and causing an estimated 290,000–650,000 deaths annually [3].

Like other RNA viruses, influenza replicates via an RNA-dependent RNA polymerase that lacks proofreading ability, leading to high mutation rates and rapid evolutionary adaptation [1,4]. As a result, RNA viruses, including influenza, exist as “quasispecies” or mutant swarms within a host. This within-host diversity, especially low-frequency mutations, is hypothesized to contribute to virulence and immune evasion [5,6,7,8]. Understanding the evolution of viruses, especially influenza, is crucial because these mutations and resultant viral diversity can significantly impact viral virulence, transmissibility, and susceptibility to immune responses, shaping the trajectory and severity of outbreaks and influencing vaccine effectiveness [5,6,7,8].

With advances in virus sequencing technologies, studying within-host viral genetic diversity has become feasible [9]. The existing studies observed relatively limited within-host genetic diversity during the acute infection [6,7,10,11,12,13,14]. Existing studies have reported relatively few variants per infection [10,11,12,13,14], suggesting that strong bottlenecks or selective pressures may limit the accumulation of within-host diversity during the acute phase of influenza infection. However, the factors that influence this diversity, especially in the context of interactions with other viruses, remain underexplored.

The motivation for this study arises from the accumulating evidence that virus–virus interactions, particularly among respiratory pathogens, can influence viral replication dynamics and host immune responses [7,15,16,17,18]. An experimental study has shown that the presence of one virus may alter the course of infection by another virus through mechanisms such as resource competition or immune modulation. For instance, a recent study showed that prior infection with the human rhinovirus (hRV) could provide protection against a subsequent influenza A virus (IAV) infection for up to three days. This prior infection reduced the IAV viral RNA by approximately 50,000-fold on day five post-hRV infection [17]. This protective effect is attributed to the hRV-induced activation of interferon-stimulated gene (ISG) expression in the airway mucosa. ISGs restrict influenza viral replication and spread, invoking immune responses in neighboring cells, thereby safeguarding them from influenza virus invasion [17,18]. Another study suggested that hRV might suppress genes linked to viral mRNA processing, ribosomal proteins, translation, and influenza infection [19]. Furthermore, a ferret model revealed that either influenza or the human respiratory syncytial virus (RSV) could induce interferon γ–expressing cells that limit the subsequent virus’s infection or disease for up to 14 days [20].

The central hypothesis underpinning our study is that viral co-infection (or co-detection) may influence the intra-host diversity of influenza by altering the selective pressures or replication dynamics of the virus. We propose that co-infection could either increase diversity by providing more opportunities for genetic variants to emerge, due to reduced competition, or decrease diversity, due to enhanced immune responses limiting viral replication.

To investigate the impact of viral co-detection on the intra-host diversity of influenza viruses, we first identified influenza viruses (both A and B) and an additional 11 respiratory viruses by applying multiplex reverse transcriptase-polymerase chain reaction (RT-PCR). Then, we performed whole-genome sequencing (WGS) of the RT-PCR-confirmed influenza cases by applying next-generation sequencing (NGS). We finally assessed the differences in the distributions of within-host minor variants (iSNVs) between the influenza-only cases and virus co-detected influenza cases.

## 2. Methods

### 2.1. Study Design

This study was a retrospective observational study of the patients presenting with influenza-like illness (ILI) symptoms during the influenza season, which was conducted October 2012 through June 2013 on a semi-isolated island population in Japan. The details of the study site were described previously [21]. Since 2007, all the ILI cases were registered and monitored as part of the national surveillance system. ILI was defined as patients presenting with a sudden onset of fever and at least one respiratory symptom of runny nose, coughing, headache, sore throat, myalgia, or fatigue. During the influenza season, all the ILI cases were tested with rapid influenza diagnostic tests (RIDT) for the influenza diagnosis. During the study period, the nasopharyngeal swabs (NPS) from the RIDTs were placed in viral transport media (VTM) and temporarily stored at −20 °C in the laboratory department of the hospital after being used for the RIDTs. Within a week, the samples were transported to the Institute of Tropical Medicine at Nagasaki University for storage in a deep freezer (−80 °C) until processing.

### 2.2. Sample Collection and Virus Identification

Viral nucleic acid was extracted using a QIA viral RNA minikit (QIAGEN Inc., Valencia, CA) following the manufacturer’s instruction. Extracted RNA samples were further processed for multiplex reverse transcriptase PCRs (RT-PCRs) to identify 13 respiratory viral pathogens: influenza A (IAV) and influenza B (IBV) viruses, human rhinovirus (hRV), respiratory syncytial virus (RSV), parainfluenza viruses 1 to 4 (PIV 1–4), human coronaviruses (hCoV) (OC43 and 229E), human metapneumovirus (hMPV), human adenoviruses (hAdV), and human bocaviruses (hBoV) using inhouse multiplex panels. The protocol of the multiplex assay panels was described elsewhere [22].

### 2.3. Library Preparation and Whole-Genome Sequencing

We amplified all 8 segments of RT-PCR-confirmed influenza samples following the protocols of Zhou, B. et al. (2009) [23] and confirmed them using gel electrophoresis. Each PCR product was purified again using Ampure XP beads (Beckman Coulter, CA, USA) according to the manufacturer’s instructions. The purity was then assessed with an Agilent Technology 2100 Bioanalyzer using a high-sensitivity DNA chip and Qubit dsDNA HS Assay Kit (Life Technology, Carlsbad, CA, USA). An amount of 1 ng of the DNA was used for library preparation (Nextera XT Kit) following the manufacturer’s instructions (Illumina, San Diego, CA, USA). After rigorous quality control, the prepared library was sequenced on the MiSeq platform (Illumina) using the V2 2 × 250 bp reagents kit.

### 2.4. Phylogenetic Analysis

Consensus nucleotide sequences for each segment, as well as concatenated WGS, were aligned using MAFFT [24]. The aligned sequences were viewed and edited in AliView [25]. Maximum likelihood phylogenetic trees were generated using IQTREE [26], with the best fit model determined by software. The non-parametric bootstrapping method was used with 100 replicates. Sequences for vaccine strains were retrieved from GISAID as outgroups. Trees were visualized and annotated using FigTree (v1.4.2) [27].

### 2.5. Identification of Intra-Host Single-Nucleotide Variants (iSNVs)

The criteria for iSNV calling were (i) mapping quality of more than 30, (ii) base quality more than 30, (iii) a minimum of a 1000-fold depth of coverage, and (iv) requiring at least 10 reads supporting an alternate allele within a single individual to evaluate the iSNV frequency. The iSNVs were included in the analysis using a cutoff frequency of 2%, following the criteria of published studies [28]. Above 50%, all single-nucleotide variants were considered to be consensus-level single-nucleotide polymorphisms (i.e., SNPs), as it is a common threshold for consensus calling in genome assembly [29].

### 2.6. Statistical Analysis

We compared the observed and expected co-detections of the viruses. First, we counted the single infections and co-infections for various combinations. Then, we calculated the expected number of co-infections in the absence of interactions between any of the pathogen pairs. This expected co-detection was calculated by multiplying the incidence of virus 1 and virus 2 by the total sample size. Then, we employed either the χ^2^ test or Fisher’s exact test (with a significance level set at *p* < 0.05) to evaluate whether there were significant differences between the observed and expected co-detections. We also computed odds ratios (ORs), along with their corresponding 95% confidence intervals (CIs), for each virus pair’s co-detection. To adjust for multiple comparisons, *p*-values were estimated using the Bonferroni and Benjamini–Hochberg methods, controlling for type I error and false discovery rates.

## 3. Results

### 3.1. Study Cohort and ILI Incidence

A total of 2407 ILI episodes were reported during the 2012/13 influenza season (from October 2012 to June 2013). Figure 1 shows the distribution of ILI cases by age group during the study period.

### 3.2. Influenza and Rhinoviruses Were Most Commonly Detected and Were a Frequently Co-Detected Pair of Viruses

Table 1 displays the characteristics of the samples collected. Out of 2396 ILI samples collected, 2316 samples were available for molecular typing of the responsible viral pathogens. The most frequently detected virus was IAV (23.2%, 538/2316), followed by hRV (11.4%, 264/2316). The other viruses detected were hAdV (1.9%,44/2316), PIV (1.9%, 43/2316), BoV (1.1%,26/2316), RSV (0.9%, 20/2316), hCoV (0.6%, 14/2316), IBV (0.9%, 22/2316), and hMPV (0.3%, 8/2316). No viruses were detected in 1427 samples. A total of 105 samples were found to have more than one virus (96 samples with two viruses, 5 samples with more than two viruses). The finding of IAV was at its peak between late February and May, and RV was observed at almost the same frequency throughout the study period (Figure 2A). hRV was the most common virus among the young age group (<3), and IAV was the most common virus among the other age groups (Figure 2B, Appendix A).

IAV and hRV were most commonly co-detected with other viruses and the most frequently co-detected pair of viruses, as shown in Table 2. However, the expected number of co-detections for RV and IAV was 61, yet 48 co-detections were actually recorded (χ^2^
*p* value ≤ 0.001), as shown in Table 3. The calculated odds ratio (OR) of rhinovirus and IAV co-detection was 0.16 (95% CI 0.09–0.28), indicating a strong negative association between the detection of IAV and RV. The other viral pairs did not have any significant difference between the observed and expected co-detections, as well as the interaction between them.

### 3.3. Whole-Genome Sequencing of RT-PCR-Confirmed Influenza A Viruses

In this study, we aimed to explore the impact of respiratory virus co-detection (or) co-infection on the IAV within-host evolution. To do so, we first sequenced the RT-PCR-confirmed influenza A viruses. We chose the IAV virus for further analysis in this study due to its dominance during the studied season. Moreover, only a very small number of IBV-positive samples were available for sequencing; thus we performed WGS for IAV-positive samples. The flowchart from sample collection to sequencing is shown in Appendix A. A total of 150 samples (only influenza = 110, and viral co-detection = 40) were successfully sequenced for the whole genome, and all were identified as A/H3N2. In Table 4, we compared the differences in demographic characteristics of influenza-only samples with viral co-detected influenza samples. A total of 60 RT-PCR positives were found to have viral co-detection; however, only 41 of these samples were available for WGS due to the low DNA concentration (Appendix A).

### 3.4. Identification of Clustering by Viral Co-Detection Status on Phylogeny

We checked whether the A/H3N2 influenza clustered phylogenetically by co-detection with another respiratory virus. If the virus–virus co-detection enhanced the antigenic variations, we would expect to see the antigenically distinct variants clustering together by the virus–influenza co-detection in phylogenetic trees. We therefore analyzed individual consensus sequences, as well as whole-genome sequences, from 132 available WGS cases (Figure 3, Appendix A). There was very little diversity among the samples, and we found that all types of virus detection (i.e., influenza alone or influenza in co-detection with other respiratory viruses) were dispersed throughout the phylogenetic tree.

### 3.5. Intra-Host Diversity in Influenza-Only Cases Versus Virus Co-Detected Cases

We next identified the iSNVs present in WGS samples. After filtering the criteria for variant calling, 132 WGS A/H3N2 samples (99 influenza-only cases and 33 co-detected cases) were retained for further analysis. A total of 397 unique minority variants were identified across the whole genome (Appendix A). Seven samples had no iSNVs identified. We did not observe samples with many iSNVs (more than 10 iSNVs/sample) or with higher frequencies of iSNVs (>10% frequencies/iSNVs), which suggests that there were few, if any, mixed-lineage infections in the samples from this season (Figure 4A, Appendix A). The average number of iSNVs per sample and the average frequency of occurrence of the iSNVs were similar across the genome, regardless of viral co-detection group (Table 5, Figure 4B). Accounting for multiple testing corrections using both Bonferroni and Benjamini–Hochberg methods, we did not identify any significant difference in the average number of iSVNs or the frequency of occurrence of iSNVs across all the segments. We did not identify any difference between influenza-only cases and virus co-detection cases by the number of days post-symptoms (Figure 4C).

### 3.6. Factors Associated with the Number of iSNVs Found in A/H3N2 WGS Samples

We identified the factors associated with a higher number of iSNVs found in A/H3N2 WGs samples (Table 6). The elderly age group (>64 years old) and school-aged children group (6–18 years old) were found to have an association with a higher number of iSNV counts (incidence rate ratio (IRR) = 1.80 (95% confidence interval (CI) 1.09–3.06), and IRR= 1.38 (95% CI: 1.01–1.90), respectively. However, after adjusting for multiple testing, the *p*-value was not statistically significant (adjusted *p*-value > 0.05). This suggests that the observed association could be due to chance rather than a true effect, and the evidence is not strong enough to rule out a random association. We did not find any other associating factors.

## 4. Discussion

Increased within-host diversity plays a vital role in viral evolution, enhancing the possibility of novel variants that natural selection can act upon [5,6]. Several factors may shape influenza A viruses’ within-host diversity and evolution. This study explores the relationship between viral co-detection and within-host influenza A virus diversity in the human host.

Many respiratory virus infections may present as influenza-like illnesses (ILI), and studies reported the presence of more than one single respiratory pathogen (viruses and bacteria) in the same patients [22,30,31,32]. From the existing studies, the virus’s co-detection rate ranges from 11.6% to 15.7% [30,33,34]. In our study, 15% (79/538) of IAV-positive samples were co-detected with other viral pathogens. We identified rhinovirus as the most commonly co-detected respiratory viral pathogen to influenza A viruses. The seasonal epidemiology of IAV and hRV may contribute to their limited co-detection, as peak incidences often occur at different times of the year. Previous studies support our findings [22,30,33,34], where hRV was reported to be the most frequently co-detected pathogen in the general population, and respiratory syncytial viruses (RSVs) were the most frequently detected among hospitalized patients [22,30,31]. None of the patients in the current study were hospitalized, which may reflect the low amount of co-detection with RSV in our study.

Based on the findings of our study, the observed negative association between IAV and RV co-detection suggests that the presence of one virus may inhibit the replication or detection of the other. Although IAV and RV were the most commonly co-detected viruses, the observed number of co-detections was significantly lower than expected (48 observed vs. 61 expected, χ^2^
*p* < 0.001). The calculated odds ratio of 0.16 (95% CI 0.09–0.28) reinforced this negative association, indicating a reduced likelihood of both viruses being detected simultaneously in the same host. Moreover, among the virus-co-detected IAV-positive samples, only 40 of the samples could proceed for the sequencing, due to the low cDNA values. These findings align with existing evidence of respiratory virus interactions, such as resources competition or induction of antiviral immune responses (e.g., ISG), leading to one virus inhibiting the infection with a second virus [16,17,18]. The experimental studies reported that the virus that infects the host cell earlier can block the entry of other viruses, or the dominant virus interferes with the replication of the co-existing viruses by triggering antiviral responses in the airway mucosa through the activation of interferon-stimulated gene (ISG) expression [16,17]. This can result in the reduction of viral RNA from a subsequent infection. If this hypothesis holds, the limited number of samples available for sequencing could be attributed to the low RNA yield in co-detected virus samples. However, we could not identify whether influenza or other viruses first infected the host in our study; thus, it was difficult to conclude. This warrants further investigation into the mechanisms underlying their interaction. The co-detections involving other respiratory viruses were infrequent, limiting the ability to draw statistically significant conclusions. This highlights the importance of increasing the sample size in future studies to better understand viral interactions and their implications for respiratory infections.

Our analysis did not find evidence to suggest that viral co-detection significantly influences the within-host diversity of influenza A. We did not find phylogenetic clustering of WGS, or individual segments based on the virus co-detection status. Moreover, the majority of cases, regardless of being solely influenza or co-detected with another virus, displayed minimal within-host minor variants, less than ten iSNVs per segment, with a frequency below 10%. Overall, the findings from our study resonate with previous studies where acute viral infections typically have limited viral diversity throughout their course [10,11,12,14,35,36].

Our study found thirteen samples with more than ten iSNV counts and ten samples with zero iSNVs. We identified that older age groups (over 64 years old) and school-aged children (6–18 years old) were more likely to have higher iSNVs than younger ones. However, in those samples with a higher number of iSNVs, the frequencies of the occurrence were less than 5%. The existing studies of within-host diversity focused on the younger age groups or immunocompromised groups; thus, it is worth further investigating to include different age groups to have better understanding of the within-host diversity of influenza viruses.

Almost 62% of the ILI samples included in this study were negative for any of the respiratory viral pathogens. The multiplex assay panels we used in the study included 13 respiratory viruses, including influenza A and B viruses. All pathogen-negative samples may be caused by bacteria or viruses that were not part of our assay panels (for example, enteroviruses or HCoV-NL63). It is worth applying more extensive multiplex assays covering the broader range of respiratory pathogens.

The study had a few limitations. First, multiple factors may influence the sensitivity and specificity of the variant calling in our study. PCR and sequencing errors may lead to the false positive variant’s calls. However, to reduce these errors, we placed the stringency criteria on the raw reads (at least 1000 depth of coverage, 100× length coverage, and variants were identified if the call was above 2% frequency). Second, our study lacked extensive longitudinal data for multiple study participants.

## 5. Conclusions

In this study cohort of 132 A/H3N2 influenza WGS sequences, we found no evidence of a specific impact of virus–virus co-detection on the within-host diversity of the influenza viral population in humans. Most of the cases, whether influenza alone or co-detected with another virus, had low iSNVs, with five iSNVs per segment and relative frequencies between 1–5%. Our findings suggest a minor or negligible contribution of these viral co-detections to the evolution of influenza viruses. However, the data available in this study may not be exhaustive, warranting further, more in-depth investigations to conclusively determine the impact of virus–virus interactions on influenza virus genetic diversity.

## Figures and Tables

**Figure 1 viruses-17-00152-f001:**
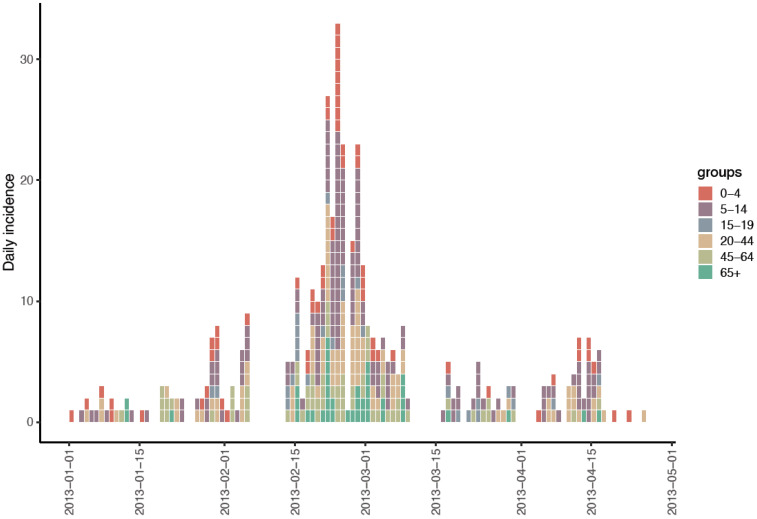
Epidemic curve of the daily number of influenza-like illness (ILI) cases reported by the hospital influenza surveillance system.

**Figure 2 viruses-17-00152-f002:**
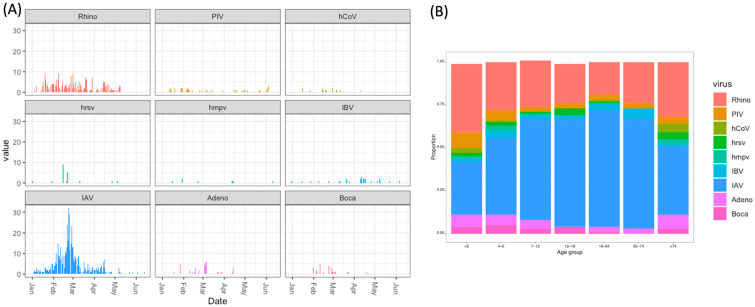
Co-circulating viruses by season (**A**) and by age group (**B**). Each color represents the individual virus detected.

**Figure 3 viruses-17-00152-f003:**
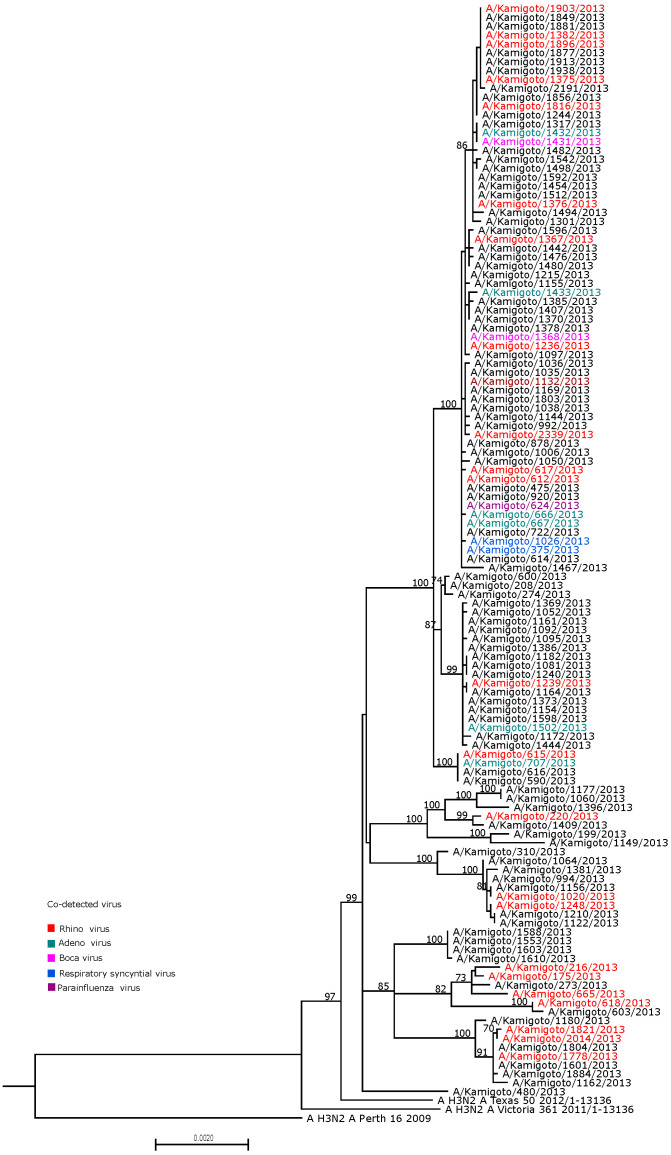
Maximum likelihood phylogenetic tree of WGS of A/H3N2 sequences in the 2012/13 influenza season. The co-detected virus, along with A/H3N2, was color-coded. Non-color-coded instances (black) are influenza-only cases.

**Figure 4 viruses-17-00152-f004:**
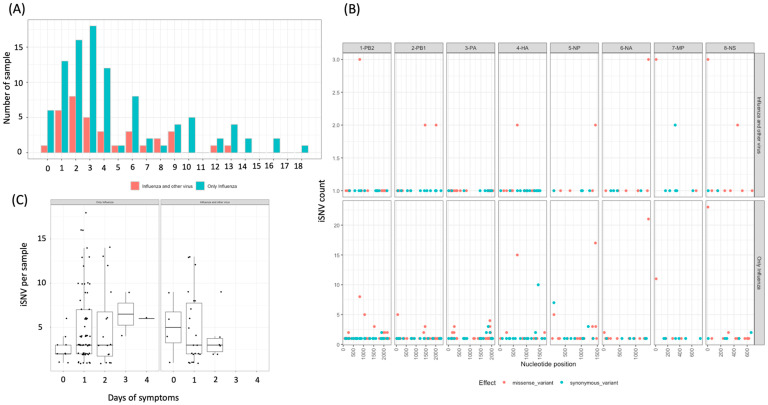
Number of iSNVs per sample (**A**), number of iSNVs by segment for influenza-only cases versus co-detected cases (**B**), number of iSNVs by days of symptoms between the groups (**C**).

**Table 1 viruses-17-00152-t001:** Characteristics of the samples analyzed in the 2012/2013 influenza season.

Characteristics	N (%)
Number of ILI episodes	2407
Age group	
<6	546 (22.7)
6–18	707 (29.3)
19–64	693 (28.8)
≥65	450 (18.7)
Total sampled	2305
By pathogen identification	
Influenza A (IAV)	538 (23.3)
Influenza B (IBV)	22 (0.9)
Human rhinovirus (hRV)	264 (11.5)
Human metapneumovirus (hMPV)	8 (0.4)
Parainfluenza virus (I-IV) (PIV I-IV)	43 (1.9)
Respiratory syncytial virus (RSV)	20 (0.9)
Human coronavirus (OC43, 229E) (hCOV)	14 (0.6)
Adenovirus (AdV)	44 (1.9)
Bocavirus (BoV)	26 (1.1)
By number of viruses detected	
Single virus infection	777 (33.7)
No virus detected	1427 (61.9)
Multiple infections	
2 viruses	96 (4.2)
>2 viruses	5 (0.2)

**Table 2 viruses-17-00152-t002:** Observed detections of single and co-infection cases among influenza-positive (538 IAV and one IBV) samples collected at the Kamigoto Hospital, Kamigoto, Japan, in the 2012/2013 season.

Virus Detected	hRV	PIV	hCoV	hMPV	RSV	IBV	AdV	BoV	IAV
hRV	264	5	0	1	0	6	8	4	48
PIV		43	0	0	0	0	0	0	4
hCoV			14	0	0	0	0	1	2
hMPV				8	0	1	0	0	0
RSV					20	1	0	0	11
IBV						22	0	0	4
AdV							44	0	7
BoV								26	3
IAV									538

Note: hRV, rhinovirus; PIV, parainfluenza virus; hCoV, human coronavirus; hMPV, metapneumovirus; RSV, respiratory syncytial virus; IBV, influenza B virus; IAV, influenza A virus; AdV, adenovirus; BoV, bocavirus.

**Table 3 viruses-17-00152-t003:** Expected versus observed co-detections in virus pairs.

Pair	Observed	Expected	Odd Ratios	*p*-Value *
hRV & PIV	5	5	0.90 (0.28–2.31)	1.000
hRV & hCoV	0	2	0.27 (0.00–2.34)	0.391
hRV & hMPV	1	1	0.97 (0.02–7.26)	1.000
hRV & RSV	0	2	0.19 (0.00–2.34)	0.156
hRV & IBV	6	3	2.09 (0.69–5.38)	0.129
hRV & AdV	8	5	1.38 (0.64–2.96)	0.544
hRV & BoV	4	3	1.18 (0.30–3.43)	0.772
hRV & IAV	48	61	0.11 (0.08–0.14)	<0.001
PIV & Adeno	0	1	0.59 (0.00–4.85)	1.000
PIV & IAV	4	10	0.31 (0.11–0.86)	0.014
hCoV & BoV	1	0	6.24 (0.14–44.04)	0.162
hCoV & IAV	2	3	0.47 (0.05–2.05)	0.389
hMPV & IBV	1	0	12.92 (0.28–103.77)	0.086
hMPV & IAV	0	2	0.19 (0.00–1.93)	0.211
RSV & IBV	1	0	5.16 (0.12–35.31)	0.190
RSV & IAV	11	5	1.79 (0.85–3.75)	0.136
IBV & IAV	4	5	0.59 (0.20–1.73)	0.484
AdV & IAV	7	10	0.51 (0.23–1.14)	0.131
BoV & IAV	3	6	0.38 (0.11–1.24)	0.121

Note: IAV, influenza A virus; RSV, respiratory syncytial virus; hMPV, human metapneumovirus; PIV, parainfluenza virus (I–IV); IBV, influenza B virus; BoV, bocavirus; AdV, adenovirus; hMPV, human metapneumovirus; hCoV, human coronavirus (OC43, 229E); hRV, rhinovirus. * Fisher’s exact test or χ^2^ test.

**Table 4 viruses-17-00152-t004:** Characteristics of the whole-genome-sequenced samples with co-detection status.

Characteristics	Only Influenza (*n* = 110)	Viral Co-Detection (*n* = 40)
Age group		
<6	8	5
6–18	41	12
19–65	52	18
≥65	6	4
Sex		
Male	52	15
Female	58	25
Vaccination status		
Vaccinated	80	26
Unvaccinated	29	14
Location/Region		
Narao	2	2
Arikawa	38	13
Shinuonome	23	4
Kamigoto	31	13
Wakamatsu	14	51
WGS (passed variant calling criteria)	99	33

**Table 5 viruses-17-00152-t005:** Average number and frequency of minor single-nucleotide variants (SNVs) detected on H3N2 sequence samples by coding region for each gene segment and viral co-detection groups.

Segment	Number of Minor iSNVs					Mean (SD) Minor iSNVs Frequency of Occurence (SD)				
	Flu Only (*n* = 99)		Flu–Other Viruses (*n* = 33)		Bonferroni-Adjusted *p*-Value	Flu Only (*n* = 99)		Flu–Other Viruses (*n* = 33)		Bonferroni-Adjusted *p*-Value
	Total Number	Mean (SD)/Subject	Total Number	Mean (SD)/Subject		Max Freq/Site	Mean (SD)/Site	Max Freq	Mean (SD)/Site	
PB2	45	1.80 (0.99)	17	1.59 (0.71)	0.698	8	0.27 (0.68)	3	0.20 (0.46)	0.700
PB1	38	1.40 (0.75)	11	2.00 (1.00)	0.192	3	0.18 (0.45)	2	0.16 (0.39)	0.980
PA	51	1.78 (1.56)	18	1.44 (0.92)	0.219	3	0.31 (0.55)	1	0.19 (0.40)	0.328
HA	34	1.59 (0.93)	13	1.85 (1.57)	0.164	15	0.18 (1.03)	2	0.18 (0.40)	0.328
NP	41	1.95 (1.30)	9	1.44 (1.01)	0.416	19	0.27 (1.57)	3	0.10 (0.39)	0.576
NA	34	1.21 (0.41)	12	1.08 (0.29)	0.545	14	0.13 (0.94)	3	0.10 (0.37)	0.663
M	19	1.05 (0.23)	9	1.22 (0.44)	0.244	11	0.07 (0.66)	3	0.08 (0.37)	0.531
NS	33	1.24 (0.50)	11	1.18 (0.41)	0.802	23	0.14 (1.35)	3	0.10 (0.39)	0.629

**Table 6 viruses-17-00152-t006:** Factors associated with the number of iSNVs found in each sequenced sample.

Characteristics	N = 132	IRR ^1^ (95% Confidence Interval)	Bonferroni-Adjusted *p*-Value
Age group			
19–64	63	1	
<6	13	0.85 (0.51–1.43)	1.000
6–18	46	1.38 (1.01–1.90)	0.280
>64	10	1.80 (1.09–3.06)	0.173
Vaccination status			
Non-vaccinated	37	1	
Vaccinated	94	0.94 (0.68–1.30)	1.000
Gender			
Female	62	1	
Male	70	0.98 (0.73–1.30)	1.000
Days post-symptoms		0.93 (0.83–1.02)	0.547
Viral co-detection			
Only influenza	99	1	
Virus co-detected	33	0.90 (0.65–1.26)	1.000

^1^ IRR = incidence rate ratio.

## Data Availability

All consensus sequences generated here have been submitted to the GISAID database. All the intra-host sequence data have been submitted to the Sequence Read Archive under BioProject, accession number PRJNA941384.

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
