# Peer review of "Impact of Viral Co-Detection on the Within-Host Viral Diversity of Influenza Patients"

_viruses, 2025, doi:10.3390/v17020152_

Round 1

Reviewer 1 Report

Comments and Suggestions for Authors Abstract: 

Line 21, iSNV has been mentioned without an explanation of its meaning.

Lines 29-20, the sentence is not clear and needs rewriting more precisely.

Expected infection value: What is the basis for calculating this value?

Author Response

Comments 1: Line 21, iSNV has been mentioned without an explanation of its meaning.

Response 1: Thank you for the comment. We have amended in the revised version.

Comments 2: Lines 29-20, the sentence is not clear and needs rewriting more precisely.

Response 2: We rephrased and upated in the revised version as “Out of the 2,305 samples tested, 538 (23.3%) were positive for influenza A virus (IAV), while rhinovirus (RV) and adenoviruses (Adv) were detected in 264 (11.5%) and 44 (1.9%) samples, respectively. Among these, co-detection of more than one virus was observed in 96 samples, and five samples showed co-detections involving more than two viruses. The most frequent viral co-detection was IAV-RV, identified in 48 out of the 96 co-detection cases.  Of the total samples, 150 were processed for whole-genome sequencing (WGS), and 132 met the criteria for intrahost single nucleotide variant (iSNV) calling. Across the genome, 397 unique iSNVs were identified, with most samples containing fewer than five iSNVs at frequencies below 10%. Seven samples had no detectable iSNVs. Notably, the majority of iSNVs (86%) were unique and rarely shared across samples. We conducted a negative binomial regression analysis to examine factors associated with the number of iSNVs detected within hosts. Two age groups—elderly individuals (>64 years old) and school-age children (6–18 years old)—were significantly associated with higher iSNV counts, with incidence rate ratios (IRR) of 1.80 (95% confidence interval [CI]: 1.09–3.06) and 1.38 (95% CI: 1.01–1.90), respectively.”

Comments 3: Expected infection value: What is the basis for calculating this value?

Response 3: The expected infection value is calculated based on the assumption of independent occurrence of the two infections. It is derived by multiplying the prevalence of each individual virus within the study population and applying this to the total sample size. For example, if Virus A is detected in 10% of samples and Virus B in 5%, the expected co-detection rate is \( 0.10 \times 0.05 = 0.005 \) (or 0.5%). Multiplying this value by the total sample size provides the expected number of co-detections.  This method assumes no interaction between the viruses, serving as a null hypothesis against which observed co-detection rates are compared to identify potential associations or interactions between pathogens.

Reviewer 2 Report

Comments and Suggestions for Authors

The topic of co-detections is quite interesting and I find the work interesting but I think that some adjustments are necessary to make the work more enlightening.

Co-detection must always be written consistently in the same format throughout the document.

Line 125: Replace hAdB with hAbV.

Expected infection value: What is the basis for calculating this value? It seems necessary to explain this analysis more thoroughly.

Age groups in Table 1: Why did you choose the specific age groups presented in Table 1? Also, why do they differ from the legend in Figure 1?

Line 108: The phrase "at least" is repeated twice unnecessarily.

Figure 1: The chart appears empty, as there is no epidemic curve shown.

Table 1:

 The age groups are incorrect, as individuals aged 65 years are excluded. The age group should be listed as "≥65".

The numbers in the table do not align:

There are 11 cases left unaccounted for in the age group data. If these cases are missing age information, this must be clearly noted.

The total sample count is listed as 2,316, but the sum of cases in the table is 2,046. Additionally, the Results section mentions 2,407 samples. These discrepancies must be clarified.

The sum of single and multiple infections (777 + 96 + 5 = 878) also requires further explanation.

The names of viruses in Table 1 differ from those in Section 2.2. Consider standardizing the format, such as using virus acronyms for readability.

Table 2: The number of PIV cases is listed as 42, whereas Table 1 mentions 43. This inconsistency needs correction.

Table 3: Again, consider using virus acronyms for ease of reading and consistency.

Influenza B: Why was Influenza B excluded from subsequent studies? This decision should be justified.

The link between the finding of the negative influence in the co-detections of Influenza virus and hRV and the final conclusion must be explained. This connection is somewhat confusing.

Author Response

For research article

Response to Reviewer  Comments

1. Summary

2. Point-by-point response to Comments and Suggestions for Authors

Comments 1: The work is very interesting, but some adjustments and corrections are necessary to make it more impactful and noticeable.

Response 1: Thank you for the comment and suggestions. We updated the manuscript accordingly to the reviewers’ comments.

Comments 2: Co-detection must always be written consistently in the same format throughout the document.

Response 2: Thank you for pointing this out. We used co-detection consistently throughout the manuscript.

Comments 3: Line 125: Replace hAdB with hAbV.

Response 3: We replaced hAdB with hAdV in the updated version.

Comments 4: Expected infection value: What is the basis for calculating this value? It seems necessary to explain this analysis more thoroughly.

Response 4: We replaced hAdB with hAdV in the updated version.

Comments 5: Age groups in Table 1: Why did you choose the specific age groups presented in Table 1? Why do they differ from the legend in Figure 1?

Response 5: The age group are now updated in the revised manuscript. The table 1 and figure 1 has same age group categories.

Comments 6: Line 108: The phrase "at least" is repeated twice unnecessarily.

Response 6: Thank you for pointing the typo. We removed the repeated phrase in the revised version.

Comments 7: Figure 1: The chart appears empty, as there is no epidemic curve shown.

Response 7: Apologies for the inconvenience. We made sure the figure is visible in the revised version.

Comments 8: Table 1:

The age groups are incorrect, as individuals aged 65 years are excluded. The age group should be listed as "≥65".

Thank you for pointing the mistakes. We have addressed the comments in the revised version.

The numbers in the table do not align:

Thank you for pointing the mistakes. We have addressed the comments in the revised version.

There are 11 cases left unaccounted for in the age group data. If these cases are missing age information, this must be clearly noted.

Thank you for the comment. The age-group is by individual headcount, ILI episodes by number of ILI episodes (some individuals have more than more ILI episodes)

The total sample count is listed as 2,316, but the sum of cases in the table is 2,046. Additionally, the Results section mentions 2,407 samples. These discrepancies must be clarified.

Thank you for pointing the mistakes. We have addressed the comments in the revised version. The sum of the cases (777+1427+96+5=2305). The updated total sample count is 2,305. The result section is also updated.

The sum of single and multiple infections (777 + 96 + 5 = 878) also requires further explanation.

We have discussed this in the results as well as discussion section as well in the revised version.

The names of viruses in Table 1 differ from those in Section 2.2. Consider standardizing the format, such as using virus acronyms for readability.

Response 8: Thank you for pointing the mistakes. We have addressed the comments in the revised version.

Table 1. Characteristics of the samples analyzed in 2012/2013 influenza season.

Characteristics

N (%)

Number of ILI episodes

2,407

Age group

   <6

546 (22.7)

  6-18

707 (29.3)

  19-64

693 (28.8)

  ≥65

450 (18.7)

Total sampled

2,305

By pathogen identification

  Influenza A (IAV)

538 (23.3)

  Influenza B (IBV)

22 (0.9)

  human Rhino virus (hRV)

264 (11.5)

  human metapneumovirus (hMPV)

8 (0.4)

  Parainfluenza virus (I-IV) (PIV I-IV)

43 (1.9)

  Respiratory syncytial virus (RSV)

20 (0.9)

  human corona virus (OC43, 229E) (hCOV)

14 (0.6)

  Adeno virus (AdV)

44 (1.9)

  Boca virus (BoV)

26 (1.1)

By number of viruses detected

 Single virus infection

777 (33.7)

 No virus detected

1,427 (61.9)

 Multiple infection

  2 viruses

96 (4.2)

  > 2 viruses

5 (0.2)

Comments 9: Table 2: The number of PIV cases is listed as 42, whereas Table 1 mentions 43. This inconsistency needs correction.

Response 9: Thank you very much for pointing out the errors. We updated in the revised manuscript. The correct number is 43 as stated in the table 1.

Comments 10: Table 3: Again, consider using virus acronyms for ease of reading and consistency.

Response 10: In the revised version, we used the virus acronyms.

Comments 11: Influenza B: Why was Influenza B excluded from subsequent studies? This decision should be justified.

Response 11: The explanation is included in the revised version. We chose the IAV virus for further analysis in this study due to its dominance during the studied season. Moreover, only a very small number of IBV positive samples available for sequencing, thus we perform WGS for IAV positive samples.”

Comments 12: The link between the finding of the negative influence in the co-detections of Influenza virus and hRV and the final conclusion must be explained. This connection is somewhat confusing.

Response 11: The link is discussed in discussion sectionBased on the findings of our study, the observed  negative association between IAV and RV co-detection suggests that the presence of one virus may inhibit the replication or detection of the other. Although IAV and RV were the most commonly co-detected viruses, the observed number of co-detections was significantly lower than expected (48 observed vs. 61 ex-pected, χ² p < 0.001). The calculated odds ratio of 0.16 (95% CI 0.09–0.28) reinforces this negative association, indicating a reduced likelihood of both viruses being detected simultaneously in the same host. Moreover, among the virus co-detected IAV positive samples, only 40 of samples could proceed for the sequencing, due to the low cDNA values. These findings align with existing evidence of respiratory virus interactions, such as resources com-petition or induction of antiviral immune responses (e.g., ISG), leading to one virus inhibiting the infection with second virus.16,17,18  The experimental studies reported the virus that infects the host cell earlier can block the entry of other viruses or the dominant virus interferes with the replication of the co-existing viruses by triggering antiviral responses in the airway mucosa through the activation of interferon-stimulated gene (ISG) expression.16,17 This can result in a reduction of viral RNA from a subsequent infection. If this hypothesis holds, the limited number of samples available for sequencing could be attributed to the low RNA yield in co-detected virus samples. However, we could not identify whether influenza or other viruses first infect the host in our study; thus, it is difficult to conclude. This warrants further investigation into the mechanisms underlying their interaction.”

For review article

Response to Reviewer  Comments

1. Summary

2. Point-by-point response to Comments and Suggestions for Authors

Comments 1: This study examines the presence of nucleotide polymorphism of influenza viruses in clinical samples taken in 2013 from patients presenting at a hospital clinic with influenza-like illness, focusing on the effects of co-infection.

Comments 2: The results showed that human rhinovirus and influenza A virus was most common, but the numbers observed were lower than might be expected if the distribution of infection was random. It was not clear to me whether the number of co-infections with other respiratory viruses were high enough for any statistical significance to be established, and this should be considered in the discussion.

Response : Thank you for the comment. We updated to include the limitation of low number of co-detection with other respiratory viruses in the discussion in the revised manuscript.

Comments 3: The authors used Rt, PCR and next generation sequencing on the Illumina platform to assess within-sample polymorphism of the influenza A(H3N2) genes comparing those from single infection to co-infection. No difference in distribution was detected.

This seems like a reasonable study, but with essentially negative results. The basis for the hypothesis – that co-infection may alter the course of an infection and so the dynamics of within-host virus polymorphism – is not unreasonable. One problem is that there might be a group of infected individuals that are the ‘selectors’ and another the ‘amplifiers’ and here the ‘selector’ group has not been sampled. 

One concern, that probably is not a problem here but needs to be considered, in studies like these is whether it is overall sequence diversity or virus diversity that is important. With such low rates of polymorphism, it seems likely that there is no sub-population of virus being detected. Moreover, the authors need to add to their assessment (e.g. in lines 357 to 363) of errors that arise during reverse transcription – these can be in the same range as or greater than an accurate PCR of of the resultant cDNA.

The authors might elaborate on the need for increased numbers and the lack of a longitudinal studies.

Response 3: Thank you for the comment. And We completely agree. We need further study including longitudinal study for the hypothesis. However, because of the funding and time limitation, we present the updated results in the manuscript, aiming to attract more funding to continue the study.

In the revised manuscript, we updated the discussion on the asssement of errors that rise during RT-PCR as well as elaborated more on the need of higher number of samples and lack of longitudinal studies.

Comments 4: There was no figure in Figure 1 in the PDF supplied by the editorial office.

Response : Updated  

Comments 5: Table 6, in its legend it is described as Table 4. It is also not clear how the presentation of the result relates to the numbers in the Table (e.g. in the text it is stated that the elderly group have an incidence rate ratio of 2.08, but in the table the number given is 1.80). 

Response 5:  We updated and corrected the typo in the revised manuscript.

Comments 6: In most cases the figure legends need to provide more detail of what is shown, but, of course, still without drawing conclusions of what is presented in the figure. Notably this applies to figures 2 and 4, and in figure 3, that the single infections were coloured in black needs to be pointed out.

Response 6: we updated in the revised manuscript

Comments 7: Abstract 

L 24, iSNV has been sued without an explanation of its meaning.

L 31-31, the sentence seems ambiguous and needs rewriting more precisely.

Response 7:

Comments 8: Introduction

L 44, the current situation with B/Yamagata lineage viruses should be mentioned.

Response: The study was retrospective 2012/13 and at the time of the project, B/Yamaga lineage viruses situation was not evolved yet, and thus we didn’t include this in the introduction.

L 59, the word ‘advancements’ should be ‘advances’.

We corrected the typo in the revised manuscript

L 72, the meaning of ‘ex vivo’ in this case is not clear and needs elaborating.

We removed “ex-vivo” in the updated manuscript.

Comments 9: Methods

L 130, the word ‘by’ is repeated.

L 154, I think the word typed as ‘singandle’ might be a typographical error.

L 159, there is a full stop/period in the line where there should be none.

L 168, a reference to Haldane’s correction should be given, the authors should describe the use of and reference the other corrections used (lines 261 and 262): the Bonferroni and Benjamini-Hochberg methods.  

Response 9: Thank you for the comment.  We corrected the mistakes in the revised version.

Comments 10: Results

L 261 and 262, see above about these corrections.

Table 5, the SNPs in M2 and NS2 need to be defined and added since they are coding regions.

Response 10: The study mainly studied the minor variants ( not SNPs) and thus we didn’t do definition of SNPs. We looked up the existing published similar studies looking at minor variants in these segments , and we followed the methodology to ensure the consistency of the reproducibility of the results.  

Reviewer 3 Report

Comments and Suggestions for Authors

This study examines the presence of nucleotide polymorphism of influenza viruses in clinical samples taken in 2013 from patients presenting at a hospital clinic with influenza-like illness, focussing on the effects of co-infection.

The results showed that human rhinovirus and influenza A virus was most common, but the numbers observed were lower than might be expected if the distribution of infection was random. It was not clear to me whether the number of co-infections with other respiratory viruses were high enough for any statistical significance to be established, and this should be considered in the discussion.

The authors used Rt, PCR and next generation sequencing on the Illumina platform to assess within-sample polymorphism of the influenza A(H3N2) genes comparing those from single infection to co-infection. No difference in distribution was detected.

This seems like a reasonable study, but with essentially negative results. The basis for the hypothesis – that co-infection may alter the course of an infection and so the dynamics of within-host virus polymorphism – is not unreasonable. One problem is that there might be a group of infected individuals that are the ‘selectors’ and another the ‘amplifiers’ and here the ‘selector’ group has not been sampled. 

One concern, that probably is not a problem here but needs to be considered, in studies like these is whether it is overall sequence diversity or virus diversity that is important. With such low rates of polymorphism, it seems likely that there is no sub-population of virus being detected. Moreover, the authors need to add to their assessment (e.g. in lines 357 to 363) of errors that arise during reverse transcription – these can be in the same range as or greater than an accurate PCR of of the resultant cDNA.

The authors might elaborate on the need for increased numbers and the lack of a longitudinal studies.

There was no figure in Figure 1 in the PDF supplied by the editorial office.

Table 6, in its legend it is described as Table 4. It is also not clear how the presentation of the result relates to the numbers in the Table (e.g. in the text it is stated that the elderly group have an incidence rate ratio of 2.08, but in the table the number given is 1.80). 

In most cases the figure legends need to provide more detail of what is shown, but, of course, still without drawing conclusions of what is presented in the figure. Notably this applies to figures 2 and 4, and in figure 3, that the single infections were coloured in black needs to be pointed out.

Minor points.

Abstract 

L 24, iSNV has been sued without an explanation of its meaning.

L 31-31, the sentence seems ambiguous and needs rewriting more precisely.

Introduction

L 44, the current situation with B/Yamagata lineage viruses should be mentioned.

L 59, the word ‘advancements’ should be ‘advances’.

L 72, the meaning of ‘ex vivo’ in this case is not clear and needs elaborating.

Methods

L 130, the word ‘by’ is repeated.

L 154, I think the word typed as ‘singandle’ might be a typographical error.

L 159, there is a full stop/period in the line where there should be none.

L 168, a reference to Haldane’s correction should be given, the authors should describe the use of and reference the other corrections used (lines 261 and 262): the Bonferroni and Benjamini-Hochberg methods.  

Results

L 261 and 262, see above about these corrections.

Table 5, the SNPs in M2 and NS2 need to be defined and added since they are coding regions.

Author Response

(The authors gave the same response as above.)

Round 2

Reviewer 1 Report

Comments and Suggestions for Authors

Thank you for addressing my comments. The manuscript is suitable for publication.

Author Response

Comments and Suggestions for Authors

Thank you for addressing my comments. The manuscript is suitable for publication.

Response: Thank you very much for the critical review 

Reviewer 2 Report

Comments and Suggestions for Authors

Thank you very much for sending the corrections and explanations. 

I have some Spelling and Formatting Suggestions:

  1. Line 179: Update the virus name to "HRV" instead of "hRV."
  2. Line 185: Correct the virus name to "RV" instead of "hRV."
  3. Table 1 and the note of Table 2: Ensure "Bocavirus" is written without a space between "Boca" and "virus."
  4. Note in Table 3: Update virus designations to align with the changes made in Table 3. The designations are currently inconsistent.
  5. Table 4: The age group should be ">=65" instead of ">65."
  6. Terminology consistency:

Use a uniform format for "co-detection" throughout the document. Ensure it is either consistently hyphenated or written as a single word ("codetection").

Similarly, ensure "coinfection" is written consistently throughout the document, either as "coinfection" or with a dash ("co-infection").

Author Response

For research article

Response to Reviewers’ Comments

Point-by-point response to Comments and Suggestions for Authors

Comments 1: Line 179: Update the virus name to "HRV" instead of "hRV."

Response 1: Thank you for pointing out the error. We updated in the revised manuscript

Comments 2: Line 185: Correct the virus name to "RV" instead of "hRV."

Response 2: We corrected in the revised manuscript

Comments 3: Table 1 and the note of Table 2: Ensure "Bocavirus" is written without a space between "Boca" and "virus."

Response 3: We corrected in the revised manuscript

Comments 4: Note in Table 3: Update virus designations to align with the changes made in Table 3. The designations are currently inconsistent.

Response 4: We updated in the revised manuscript

Comments 5: Table 4: The age group should be ">=65" instead of ">65."

Response 5: We corrected in the revised manuscript

Comments 6: Terminology consistency:

Use a uniform format for "co-detection" throughout the document. Ensure it is either consistently hyphenated or written as a single word ("codetection").

Similarly, ensure "coinfection" is written consistently throughout the document, either as "coinfection" or with a dash ("co-infection").

Response 6: We used co-detection and co-infection consistently in the revised manuscript